# A Combined Field and Remote-Sensing Based Methodology to Assess the Ecosystem Service Potential of Urban Rivers in Developing Countries

**Manuel R. Beißler [1] and Jochen Hack [2,*]**

[1]   Technische Universität Darmstadt, Department of Civil and Environmental Engineering, Franziska-Braun-Str. 3, 64287 Darmstadt, Germany

[2]   Technische Universität Darmstadt, Institute of Applied Geosciences, Section of Ecological Engineering, Research Group SEE-URBAN-WATER, Schnittspahnstr. 9, 64287 Darmstadt, Germany

*   Correspondence: contact@geo.tu-darmstadt.de; Tel.: +49-6151-162-0981

**Abstract:** Natural rivers in urban areas bear significant potential to provide ecosystem services for the surrounding inhabitants. However, surface sealing by houses and street networks, urban drainage, disposal of waste and wastewater resulting from advancing urbanization usually lead to the deterioration of urban rivers and their riparian areas. This ultimately damages their ability to provide ecosystem services. This paper presents an innovative methodology for a rapid and low-cost assessment of the ecological status of urban rivers and riparian areas in developing countries under data scarce conditions. The methodology uses a combination of field data and freely available high-resolution satellite images to assess three ecological status categories: river hydromorphology, water quality, and riparian land cover. The focus here is on the assessment of proxies for biophysical structures and processes representing ecological functioning that enable urban rivers and riparian areas to provide ecosystem services. These proxies represent a combination of remote sensing land cover- and field-based indicators. Finally, the three ecological status categories are combined to quantify the potential of different river sections to provide regulating ecosystem services. The development and application of the methodology is demonstrated and visualized for each 100 m section of the Pochote River in the City of León, Nicaragua. This spatially distributed information of the ecosystem service potential of individual sections of the urban river and riparian areas can serve as important information for decision making regarding the protection, future use, and city development of these areas, as well as the targeted and tailor-made development of nature-based solutions such as green infrastructure.

**Keywords:** Urban rivers; ecological status; ecosystem services; developing countries; Nicaragua; nature-based solutions; green infrastructure; MAPURES; matrix approach

## 1. Introduction

Ecosystem services are the direct and indirect benefits that people derive from various types of ecosystems [1]. The maintenance and enhancement of ecosystem services is of central interest to societies. According to Potschin and Haines-Young [2], specific ecological and biophysical structures and processes, representing the status of an ecosystem (natural capital), can be conceptually linked to elements of human well-being [3]. A good ecosystem status provides adequate biophysical structures (e.g., for habitats) and facilitates ecological processes (e.g., primary production), while a deteriorated ecosystem status has less potential to provide such functions that ultimately lead to benefits to human beings as ecosystem services.

Recently, there has been increasing scholarly interest (e.g., several special issues of international journals) in the area of ecology and ecosystem services provided by urban areas [4–6]. Due to the high population density of urban areas, there is potentially a high demand for ecosystem services, especially those provided in situ [7]. However, urban areas have only recently been considered as potential provisioning areas of ecosystem services [8,9]. Many studies have focused either on, urban green spaces [10], or hydrological ecosystem services of river basins [11]. In contrast, urban rivers and their potential to provide in situ ecosystem services have not been considered to the same extent. Francis (2012; [12]) provides an overview of research on urban rivers highlighting a strong increase in publications on the subject from the 2000s on. However, a large majority deals with urban rivers in industrialized countries and less than 1% addresses ecosystem services of urban rivers. This misses the fact that the potential of rivers and riparian areas in good ecological status to provide ecosystem services, due to their diversity in aquatic and terrestrial biocenosis and biotopes, is much higher compared to other ecosystems [13–15].

Although people in developing countries rely more directly on ecosystem services than people in developed countries [16], they are often the ones who degrade ecosystem services they depend on [17]. Thus, urban rivers being largely used for disposal of untreated waste and storm water discharge, as well as local dump sites [18,19]. Nevertheless, the restoration of the physical conditions, and thereby the improvement of ecosystem services, can be worthwhile even from a purely economic point of view [15], especially when the potential benefits of urban rivers are manifold [20]. However, urban rivers in developing countries are still in relatively good physical condition. For example, hydromorphological changes such as channelization or artificial embankments of small and medium-size rivers are often not as advanced as in developed countries [21]. Especially in small- and medium-sized towns, the river morphology and often also the riparian vegetation is in relatively good condition. With increasing urbanization and often-unguided urban development, however, the ecological potential of urban rivers is at increased risk of being lost [22]. Moreover, urban rivers and riparian areas are often encountered in urban-rural transitioning zones representing places of informal settlements. This can be explained because these areas represent still somehow rural living conditions, to satisfy basic needs for water and waste disposal, and offer the potential for agriculture for marginalized city dwellers. Since these areas are often prone to flooding and contamination, they are not priority areas for formal city development but are rather left aside [23].

The concept of ecosystem services encourages problem solving by highlighting the social benefits that intact water bodies and riparian areas can provide [24]. By identifying the potential for ecosystem services, it is assumed that urban water pollution could be reduced and urban rivers in developing and emerging countries could be maintained in more natural state [25]. However, there is no widely accepted methodology for how to evaluate the value of urban rivers and riparian areas in their ability to provide ecosystem services [26].

This publication presents a novel methodology, MAPURES V1.0 (Methodology to Assess the Potential of Urban River Ecosystem Services—Technische Universität Darmstadt, Darmstadt, Germany), to assess the ecological status of urban rivers and riparian areas in developing countries under data scarce conditions in order to derive the potential to provide ecosystem services according to the internationally recognized Common International Classification of Ecosystem Services (CICES) [27]. The developed methodology addresses the need for tools to map and quantify spatial explicitly ecosystem services provided by urban rivers [28]. Based on easily obtainable low-cost field data and freely available high-resolution satellite images three ecological status categories: hydromorphology, river water quality, and riparian land cover are evaluated and related to Urban River Ecosystem Service (URES) using a weighted indicator-based matrix approach. Matrix approaches are widely used to relate ecological indicators such as land cover or other proxies for biophysical structures with the provision potential of ecosystem services [8,29–37]. MAPURES V1.0 is being developed in the context of the SEE-URBAN-WATER project (www.tu-darmstadt.de/see-urban-water) as a guiding tool for municipalities in Nicaragua and Costa Rica to highlight the provision of ecosystem services of

urban rivers in order to promote their protection. The specific novelty of our approach is the explicit assessment of URES through a combined use of high-resolution remote sensing and field-based data that comprehensively covers different biotic and abiotic realms of rivers, as well as the land cover characteristics of the river corridor. The resulting spatially explicit assessment of URES linked to different ecosystem status categories is new. As a result of applying our approach, important new knowledge is gained regarding the potential benefits urban rivers can provide and whether their provision is endangered or not.

This universal methodology is applied to the case study of the Pochote River in crossing the city of León, Nicaragua, for the regulating ecosystem services of this urban river and its riparian areas. As a result, the potential to provide each identified URES for every 100 m section of the considered river is quantified and can be visualized via geo-referencing. This spatially distributed information of URES potential of individual sections of the urban river and riparian areas can serve as important information for decision making regarding the protection, future use, and city development of these areas as well as the targeted and tailor-made development of nature-based solutions such as green infrastructure. The methodology presented in this publication also supports the rapid appraisal of URES, which can be applied by untrained professionals and with low resource investment. The study represents a first practical application of the CICES framework for an urban river ecosystem in a data-scarce region.

## 2. Materials and Methods

### 2.1. Study Area and Raw Data

The Pochote River has its origin in the northern part and limits of the city of León. With about 180,000 inhabitants, León is Nicaragua's second-largest city, located about 93 km northwest of the country's capital Managua [38]. While this origin area extends about 20 km to the North-East of the City, the river flows in South-West direction along the city boundary for about 6 km. A few kilometers after leaving the city limits, it confluences with the Chiquito River, another river of the City of León, forming a common river basin that leads to the Pacific Ocean. The climate of the study area (Figure 1) is a tropical savannah climate with a pronounced dry season from November to April and a rainy season from May to October. The average monthly precipitation ranges from 300 to 500 mm. The average daily temperature varies from 27 to 29 °C, with the lowest values found between the months of December and February [39].

The current state of these water bodies is characterized by high levels of contamination due to the disposal of both domestic waste and waste water as well as industrial effluents stemming from the nearby tanneries and slaughterhouses. The Pochote River has recently become part of the urbanized area due to rapid urban expansion throughout the last decades.

The origin of the Pochote River inside the city is comprised of three channels, each having an approximated length of 1 km. Although they are located within a highly urbanized area, the access to the river is partly limited due to a canyon-like topography. On the mid-course, after the three channels have merged, a single stream channel forms meanders and floodplains. These hydro-morphological characteristics have led to different natural conditions surrounding the river and also influenced different formal and informal settlement patterns. As indicated in Figure 1, the settlement diversity along the river shores changes in flow direction from formally and densely urbanized (1) to informal urban-rural transitional (2) and to formal rural settlements (3). Due to this urban and ecological diversity, a highly fragmented landscape exists. The most relevant water-related issues are a consequence of the inefficient operation of the local wastewater treatment plants and sewerage system as well as direct wastewater discharge from households alongshore, hydraulic stress due to surface runoff from sealed surfaces, and uncontrolled garbage burning and disposal [40].

Additionally, the clearing of riparian vegetation is increasingly a problem disturbing the ecology of the river. However, residents, especially those in more rural settlements along the river, use the river water and that of the numerous natural wells in short distance to the river course in their daily lives

(e.g., for household duties and livestock needs) [40]. The natural wells deliver significant amounts of clear water to the river even during the dry season when the river is not fed by rainfall at all but only waste water from households. The contribution from natural wells leads to an important dilution of waste water and, together with the natural morphology of the river, results in a continuous improvement of the water quality downstream. This process is regarded as a natural cleaning. It is assumed that a constant natural biodegradation and phytoremediation of (mostly) organic pollutants of discharged household wastewaters occurs when the river water flows downstream. Based on expert insights from different field visits to the Pochote River and the surroundings of León, a flow distance of 10 km was assumed to be necessary to naturally improve the water quality from category 4 (highly polluted) to category 0 (not or barely polluted) if there was no further contamination along this flow distance. This means water quality improved in one category per 2.5 km. Such a water quality improvement of one category could be observed at the western end of the study area over a flow distance of 2.5 km [40]. During a field survey from May to June 2017, the exact course of the Pochote River, as well as geolocations of specific points of interest: solid waste disposal sites, waste water discharges, natural springs and tributaries as well as any constructive modification of the hydromorphology of the river channel (e.g., bridges, weirs, artificial embankments), were georeferenced and recorded. Additionally, a georeferenced photograph was taken for all points of interests, described and documented in an exhaustive photo documentation (see Figure A1 in the Appendix A, and [40]). The points of interest identified in May and June 2017 were validated during several other field surveys between September 2017 and April 2018. Field surveys conducted during both rainy (May–October) and dry season (November–April) revealed additional insights in the seasonality of river flow (e.g., in order to distinguish perennial and ephemeral streams) and permanence of sources of contamination (e.g., the discharge of untreated wastewater) throughout the year. The information gathered during these field survey represents the 'raw data' for the assessment of the ecological status of the Pochote River regarding hydromorphological quality and water quality. However, the data collection did not follow a predefined pattern (e.g., every 100 m along the river; see Methodology section) since the objective of the field surveys was to geo-reference the river network, points of contamination, and specific characteristics of river morphology.

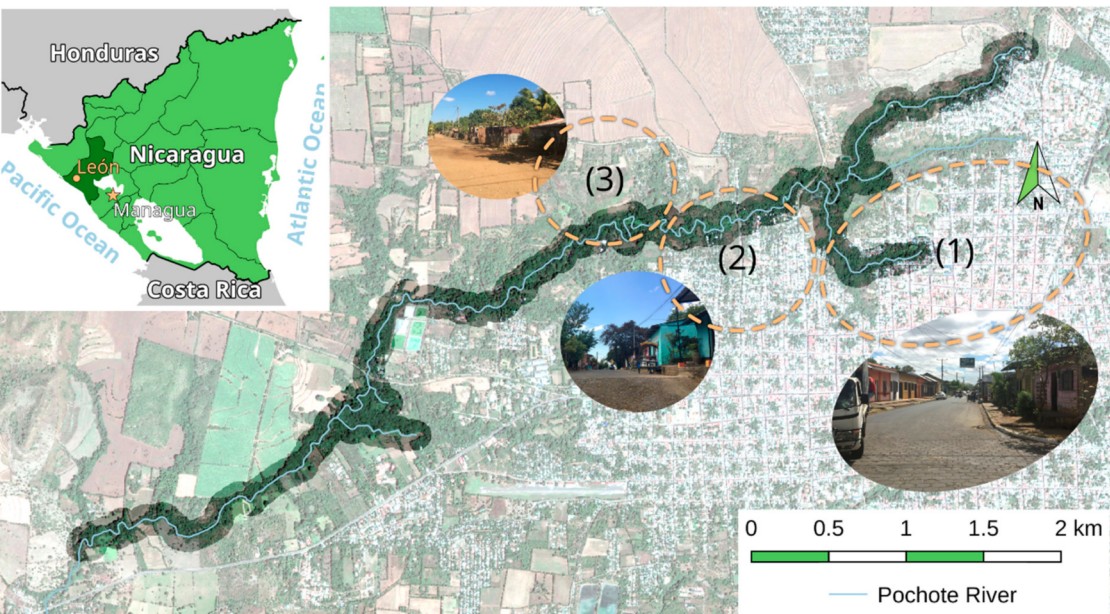

**Figure 1.** Overview of study area: Pochote River and City of León with different settlement structures: (1) formal and densely urbanized, (2) informal urban-rural transitional and (3) formal rural settlements.

## 2.2. Methodology

The methodology described in this section was developed to be easily adaptable to the data scarcity in the project region. Such data scarcity is typical in developing countries. The proposed methodology and data used allows an easy and low-cost application. However, the methodology can also be applied with other kind of data, e.g., more quantitative and detailed remotely sensed or field data. Figure 2 illustrates the individual steps of the methodology.

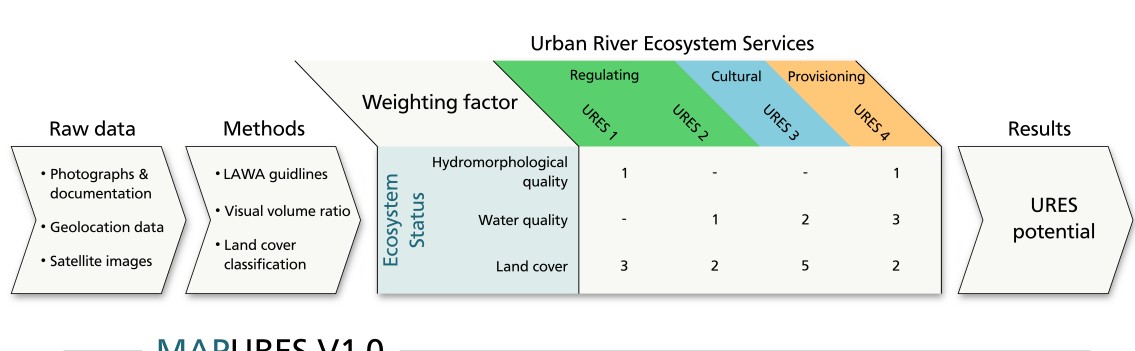

**Figure 2.** Methodology MAPURES V1.0 to assess the ecological status and the potential to provide ecosystem services of urban rivers.

The framework is based on the assessment of three ecosystem status categories: 'water quality', 'hydromorphology', and 'land cover'. These categories were chosen because they comprehensively cover different biotic and abiotic realms as well as the characteristics of a river corridor. The category 'water quality' provides information on the biotic conditions of the aquatic environment. It can be used to qualitatively assess the provision of ecosystem services related to aquatic life in a coherent manner. The ecosystem status category 'hydromorphology' covers the dynamics of the abiotic realm of the river corridor. It serves to assess the diversity of habitats and habitat conditions. Important natural hydrodynamics, as well as habitat-changing or -dividing (man-made) disturbances of them (e.g., contouring or transversal structures such as artificial embankments or weirs), as well as the transition of river bed to riparian zone (i.e., aquatic to terrestrial habitats are considered within this category). Finally, the 'land cover' category reflects the conditions of the riparian vegetation with the wider river corridor (200 m wide strip). Since, a characterization of vegetative cover of the river corridor (ca. 400,000 m$^2$) in ground work is very labor intensive, a characterization via high-resolution satellite images are used in our methodology to efficiently produce a land cover map.

The ecosystem status is then examined for links to Urban River Ecosystem Services, using the indicators of Maes et al. [34,35] to assess specific URES for the (new) URES biome-related category. In the following, the links of ecosystem status categories to individual URES are defined as weighted multi-criteria equations and used to assess the URES potential of the Pochote River.

In the following, each methodological step from the assessment of ecosystem status over the establishment of a URES categorization to the examination of URES potential is explained in more detail.

### 2.2.1. Hydromorphological Quality

To determine the hydromorphological quality of individual river sections, a method of the German state working committee LAWA [41] was used in combination with the geo-referenced photographic documentation and photographs from complementary field surveys. The LAWA method is intended for mapping the hydromorphological quality of small- and medium-sized running waters. It has been successfully applied for many rivers in Germany [42]. However, due to the lack of data for the study area, the following subcategories could not be considered: 'Longitudinal banks', 'Special river course structures', 'Transversal banks', 'Flow diversity', 'Depth variance', 'Substrate diversity', 'Special

river bed structure', 'Special bank structures)'. In addition, the category 'River environment' was not considered, in order to avoid duplication with the land cover assessment. Thus, with the exception of this category, 14 out of 22 subcategories of the LAWA method [41] were considered. The geographic information system QGIS (2.18 LTS, 2016-10 [43]) was used to illustrate the hydromorphological quality based on the LAWA quality classification. Since the river is not wider than 10 m over its whole course, the river was divided into 100 m sections according to the LAWA guidelines (a total of 121 sections). The coloring was also carried out on the basis of these guidelines according to the different hydromorphological quality classes: 1 = 'unchanged', 2 = 'slightly changed', 3 = 'moderately changed', 4 'significantly changed', 5 = 'strongly changed', 6 = 'very strongly changed', 7 = 'totally changed'. For the sections in which no photos and no specific local knowledge existed (48 sections), a linear adjustment of the previous and the following section was assumed.

### 2.2.2. Water Quality

To assess the ecosystem status regarding water quality, a qualitative map with the sources of contamination was created using the available geo-referenced photo documentation [40] and additional site visits. This qualitative evaluation was mainly a visual analysis of the photographs and secondarily based on site descriptions of the geo-referenced photo documentation. For water quality rating, a categorization was generated: category 0 'barely or not polluted' for visibly clear water, up to category 4 'highly polluted' for milky whitish colored water and/or foaming water. The addition 'seasonally' was used if there were discharges in photos in the rainy season, but absent in photos of the dry season or if the photos gave cause for this addition for other reasons. The classes which were created for the classification can be seen in Table 1. The map resulting from this processing is shown in Figure 3.

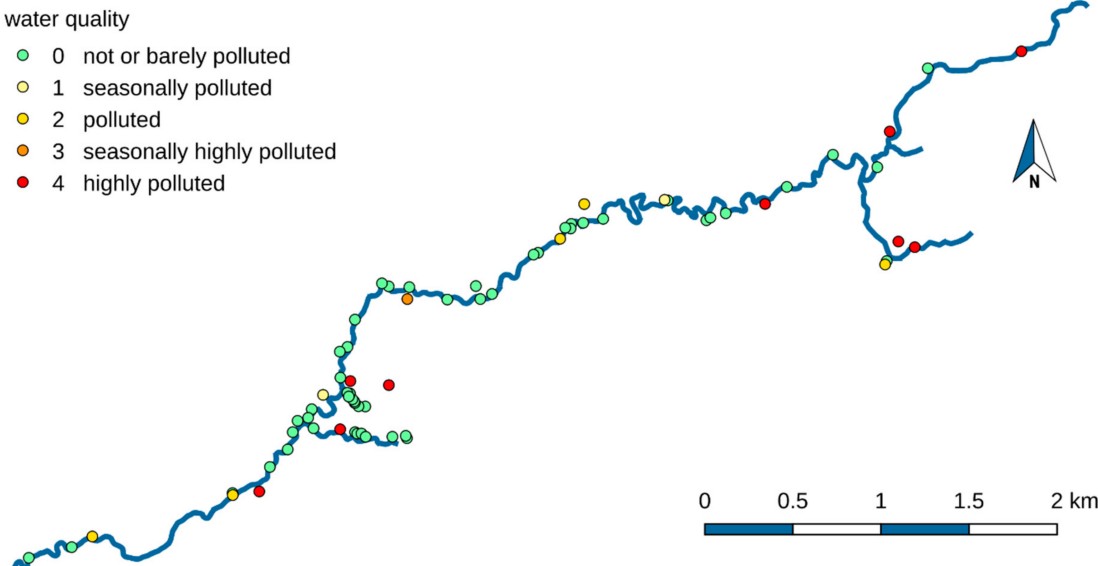

**Figure 3.** Location of photographs indicating degree of pollution of the Pochote River based on field survey information [40].

**Table 1.** Classification of water quality; Author's work.

| # | Class | Appearance of Discharge |
|---|-------|-------------------------|
| 0 | not or barely polluted | visibly clear water |
| 1 | seasonally polluted | brownish colored water, rainwater from surface runoff or does not fit in the other categories |
| 2 | polluted | |
| 3 | seasonally highly polluted | milky whitish colored water and or foaming water |
| 4 | highly polluted | |

To produce a map for the water quality from this map, the same 100 m sections that had already been introduced for the hydromorphological quality classification were used. The classification of contamination was retained for water quality. The water quality of individual river branches contributing to the main river was assumed to be in class 1 (seasonally polluted) in the urban regions and class 0 (not or barely polluted) in the non-urban regions. This is based on the assumption that the proportion of urban surface water in the urban spring regions is likely to be relatively high—and thus of poorer quality—due to surface sealing and in the non-urban spring regions it mainly consists of groundwater, which is assumed to be of better quality. Since no quantitative discharge data were available, the water quality at points of confluence of river branches and the main river was calculated using the volume ratio (*R*) between the (visual) discharge of the main river section and the discharge of a tributary or an affluent. This ratio was determined visually on the basis of photographs at 4 confluences of river branches, 59 spring water inflows (water quality class 0 in Figure 3) and 17 waste water discharges (water quality classes 1–4 in Figure 3). The visual assessment of water quality applied here is similar to the Secchi Disc method (Forel-Ule scale) [44,45] and black disc method [46,47], both common methods to visually determine surface water transparency and quality, the visual estimation of the relative ratio between river and tributary flow was based on expert knowledge of typical flows by given cross-sections. The pollution class for the river section after the confluence is calculated depending on the volume ratio and the pollution classes. To simplify the calculations, values between 0 and 1 were assigned to the individual water quality classes (Table 2). Additionally, a natural cleaning factor (*N*) was also assumed to clean the river from category 4 to category 0 in 10 km. The calculations were carried out starting at the sources of each river branch (source areas) that contributes to the main river. For each confluence with an affluent or a tributary, the water quality value of the river section was estimated using the aforementioned visual volume ratio (*R*) of the discharge to the previous section. This procedure is described below in formula (1) and (2).

$$WQ_{nc,j} = \left(1 - R_{tr,j}\right) \cdot \left(WQ_{nc,j-1} \cdot \left(1 - \sum_i R_{di,i,j}\right) + \sum \left(WQ_{di,i,j} \cdot R_{di,i,j}\right) + WQ_{tr,j} \cdot R_{tr,j}\right) \tag{1}$$

$i$ = Number of discharge in one section
$j$ = Number of section
$WQ_{nc}$ = Water quality value without applied cleaning factor (-)
$WQ_{di}$ = Water quality value of discharge (-)
$WQ_{tr}$ = Water quality value of tributary (-)
$R_{tr}$ = Visual volume ratio between the tributary and the main river (-)
$R_{di}$ = Visual volume ratio between the discharge and the tributary/main river (-)

$$WQ_j = WQ_{nc,j} + N * \frac{l}{100} \tag{2}$$

$WQ$ = Water quality value (with applied cleaning factor) (-)
$N$ = Natural cleaning factor (1/m)
$l$ = Section length (m)

In addition, an upper limit for the water quality value of 1 was introduced as the quality cannot increase above this optimal value. After all calculations, the values were then assigned to the water quality classes (see Table 1) using class limit values.

### 2.2.3. Land Cover

A land cover analysis of 100 m buffer strips on both sides of the river (study area; see Figure 1) was carried out in order to assess the vegetative status of the riparian corridor of the Pochote River. To be able to analyze even very small-scale structures and patches, which are very common in urban river environments, high-resolution satellite images provided by Google Earth were used. An innovative methodology for land cover classification on the basis of these available, which are freely available for non-commercial uses, was developed (see Figure A2 in the Appendix A).

First of all, an orthogonal bird's-eye view image of the study area was generated using Google Earth (Google LLC, Mountain View, CA, USA—Version: 7.3 Desktop) and exported in jpg-format. The chosen image was taken during the dry season, specifically on 26 January 2017 (Digital Globe, GeoEye1–Digital Globe, Denver, CO, USA). The advantage provided by using an image during dry season is that is shows a greater contrast between the evergreen trees and other perennial vegetation and agriculture areas, making it easier to discriminate between these types of vegetation. In addition, the image is relatively up-to-date. The image was then edited for land cover classification with the GNU Image Manipulation Program (GIMP) (Version 2.8, 2017-05, GIMP Development Team [48]), a widely used software program for image editing. At the beginning, the satellite image was imported into GIMP and roughly cropped to the study area. Then the 'Select by Color Tool' (GIMP Development Team) was used to select and colorize one area type after the other. After that small adjustments were necessary i.e., to have only three different colors in the selected area. This serves for easier handling and error minimization in QGIS. For the same purpose, the color palette was reduced to only three colors before being exported as a .png file with the '8bpc GRAY' (GIMP Development Team) pixel format. Then the original satellite image was opened with the 'Georeferencer GDAL' plugin (QGIS Development Team) and geo-referencing points were determined using the 'OpenLayers Plugin' (QGIS Development Team) [49]. These geo-referencing points were used to geo-reference (WGS84 ESPG: 4326) the produced map (exported .png file) to obtain the same dimensions as the original satellite image. The resulting raster was finally cropped to the study area (with a 100 m buffer zone on both sides of the river) and also divided into 100 m sections in QGIS. For this purpose, the layer of the study area had to be divided into the same 100 m sections as the other two ecosystem status categories. Then the 100 m section layer could be cut with the raster to achieve the desired result. The resulting land cover map representing, in addition to Hydromorphological and water quality, the third ecosystem status category used to derive the ecosystem service potential of the Pochote River. To use the land cover classes as input data for the assessment of ecosystem service potential, the following unified values are assigned to the three land use classes 'high and perennial vegetation' = 1.0, 'low vegetation' = 0.6. and the 'built-up' = 0. These values are based on the assumption that 'built-up' area has no potential to provide ecosystem services at all (value 0), while 'high and perennial vegetation' has the full potential (value 1.0) to provide ecosystem services. 'Low vegetation' (i.e., seasonally disappearing) is considered to be disturbed vegetation that can only provide a certain degree (here 0.6) of the total potential of ecosystem services. The value 0.6 was chosen to indicate that the potential of 'low vegetation' is still closer to the potential of 'high vegetation' than to that of the 'built-up' area.

Figure A2 in the Appendix A contains a detailed step-by-step description of the work flow that was applied to reach the land cover classification for the study area.

### 2.2.4. Urban River Ecosystem Services Categorization

An identification and categorization for URES is required since this category of ES has not been explicitly defined yet. This newly defined category of URES represents a urban terrestrial aquatic specific selection of ES classes defined in the Common International Classification of Ecosystem

Services (CICES) [27]. To generate this new categorization, categorizations for urban ES and river ES classes (as part of the fresh water ES categorization of Maes et al. [34,35]) were combined, except those ES classes with groundwater relation. Urban river ecosystems are also linked to the non-urban river ecosystems of the river by dynamic abiotic processes. In addition, urban ecosystems can be located anywhere in the city, including along the river (e.g., park trees and allotments). The idea of a simple combination of river ecosystem services and urban ecosystem services is based on these two circumstances. Groundwater-related services were not considered explicitly, because they are not considered part of river ecosystem services, although groundwater use by bank filtration is possible [2]. Only those services for which indicators were available have been selected, to ensure unambiguity, consistency, and ease of use as indicators do not need to be developed. This selection has been adapted to the latest CICES version (CICES V5.1) using the CICES spreadsheet [50] representing the URES categorization referred to in this publication (Table S1 in the Supplementary Materials).

### 2.2.5. Links between URES and Ecosystem Status Categories

To calculate the potential of urban rivers to provide ecosystem services on the basis of ecosystem status categories using the previously determined conditions for each of the Urban River Ecosystem Services (URES), it was necessary to normalize the different categories. As with the water quality category, values between 0 and 1 were chosen (Table 2), allowing the results of the status to be expressed in percentage.

**Table 2.** Transformation of classes in normalized values.

| Hydromorphological Quality | | Water Quality | | Land Cover | |
|---|---|---|---|---|---|
| Class | Value | Class | Value | Class | Value |
| 1 unchanged | 1.0 | 0 not or barely polluted | 1.0 | high vegetation | 1.0 |
| 2 slightly changed | 0.83 | 1 seasonally polluted | 0.9 | | |
| 3 moderately changed | 0.67 | 2 polluted | 0.6 | low vegetation | 0.6 |
| 4 significantly changed | 0.50 | | | | |
| 5 strongly changed | 0.33 | 3 seasonally highly polluted | 0.3 | | |
| 6 very strongly changed | 0.17 | | | | |
| 7 totally changed | 0.0 | 4 highly polluted | 0.0 | built-up | 0.0 |

Since the water quality contains a constant cleaning factor instead of a value based on the heterogeneous hydromorphology and since these two categories are spatially strongly separated from the land cover, it is assumed that all the categories are only slightly interdependent. This independence given by the survey methods allowed for development of a simple matrix linking ecosystem status categories with URES via weighting factors (Table 3). The assessment of ecosystem services based on land cover respectively habitats by matrices is a commonly used method [51–53]. New in this work is that the matrix from Table 3 translates not only land cover but also water quality and hydromorphological quality into the potential to provide ecosystem services. The weighting factors are based on the combination of URES indicators (Table S1; originated from Maes et al. [34,35]) and findings from site visits, expert opinions and the experiences of knowledgeable local residents. The scarce data did not allow making appropriate statements about some of the initially selected URES that is why some in the table were greyed out. The decision on how the general development of the weighting factors is to be determined is briefly explained below using individual ecosystem services as examples. According to Table S1 the same indicators are commonly used for the services 'Bio-remediation by micro-organisms, algae, plants, and animals' and 'Filtration/sequestration/storage/accumulation by micro-organisms, algae, plants, and animals', ranging from indicators of the general ecological status and specific indicators of water quality and to the area occupied by riparian forests. Therefore, all three

ecosystem status categories can be assigned to these two URES. Since all three also link to other different indicators (i.e., all cover their own area, they are added to one third each).

Since no temperature values were measured, only the 'Leaf Area Index' can be considered for the 'Regulation of temperature and humidity, including ventilation and transpiration' service (Table S1). Therefore, land cover is linked to this service as it is very similar to the Leaf Area Index.

There is a whole range of indicators for the 'Characteristics of living systems that enable activities promoting health, recuperation or enjoyment through passive or observational interactions' services. The only indicator related to the ecosystem status categories considered here is 'bathing water quality', which could be determined by water quality. However, since there are so many various indicators here, it makes little sense to use water quality alone. Hence, this URES class will not be considered in this study.

Regarding the 'Characteristics of living systems that enable aesthetic experiences' service, surveys of the population on how beautiful they find the surroundings around the Pochote River have yet to be collected.

**Table 3.** Relations between the URES and ecosystem status data.

| Urban River Ecosystem Service (URES$_k$) | | Weighting Factors of Ecosystem Status Categories | | |
|---|---|---|---|---|
| | | Hydromorphological Quality ($hq_k$) | Water Quality ($wq_k$) | Land Cover ($lc_k$) |
| **Regulating ES** | Bio-remediation by micro-organisms, algae, plants, and animals | 1 | 1 | 1 |
| | Filtration/sequestration/storage/accumulation by micro-organisms, algae, plants, and animals | 1 | 1 | 1 |
| | Noise attenuation | - | - | 1 |
| | Hydrological cycle and water flow regulation (Including flood control, and coastal protection) | - | - | 1 |
| | Maintaining nursery populations and habitats (Including gene pool protection) | 1 | 1 | 2 |
| | Regulation of the chemical condition of freshwaters by living processes | 3 | 1 | 1 |
| | Regulation of temperature and humidity, including ventilation and transpiration | - | - | 1 |
| **Cultural ES** | Characteristics of living systems that enable activities promoting health, recuperation, or enjoyment through passive or observational interactions | - | - | - |
| | Characteristics of living systems that enable aesthetic experiences | - | - | - |
| **Provisioning ES** | Surface water for drinking | - | - | - |
| | Surface water for non-drinking purposes | - | - | - |

As can be seen in the table, indicators for provisioning and cultural URES could not be directly linked to the ecosystem status categories considered in this study and are therefore neglected.

The following formula shows how URES for each 100 m river section are calculated based on a weighted contribution of values for the three ecosystem status categories:

$$URES_{j,k} = \frac{HQ_j \cdot hq_k + WQ_j \cdot wq_k + LC_j \cdot lc_k}{hq_k + wq_k + lc_k} \qquad (3)$$

$k$ = Type of URES

*j* = Number of section
*URES* = Urban River Ecosystem Service (-)
*HQ* = Hydromorphological quality value (-)
*hq* = Hydromorphological quality weighting factor (-)
*WQ* = Water quality value (-)
*wq* = Water quality weighting factor (-)
*LC* = Land cover value (-)
*lc* = Land cover weighting factor (-)

## 3. Results

The application of the methodology described in the preceding section produced three ecosystem status maps (hydromorphological status, water quality and land cover), which are intermediate results, and potential URES maps, one for each of the ecosystem service assessed, which are final results.

### 3.1. Ecosystem Status Categories

For the hydromorphological quality (top image in Figure 4), the river can be classified as good almost over its entire length, as the hydromorphological quality rarely goes beyond class 3, with the exception of the densely populated area in the northeast and due to a few individual structural elements along the river. A strongly meandering section at the middle course of the river as well as the last tributary (stemming from the Botanic Garden) that confluences with the main river in the southwest are classified as "unchanged", reflecting the highest potential to provided hydromorphology related URES.

The central image of Figure 4 shows, based on the methodology presented in Section 2.2.2, the water quality per 100 m river section in the study area. Similar to the hydromorphological quality, it is significantly worse in the densely populated area in the northeast. The presence of several highly polluted (classes 3 and 4) river sections within this area, and better water quality classes farther away from it, corresponds well to the empirical evidence from field surveys and seems logical. Abrupt improvements from one river section to the following one are a result from a high number of fresh water sources (natural wells) that reduce the concentration of contaminants due to dilution. From the strongly meandering middle course of the river on several natural wells contribute (apparently) clean water to the main river. This correlates with an increasing distance of the river to the urbanized area when flowing downstream. The downstream trend in cleaner river sections can be explained by the self-purification of the river, assumed to be natural cleaning, as explained in Section 2.1, and the continuous contribution of discharge from natural wells.

The bottom image in Figure 4 illustrates the land cover for each 100 m long and 100 m wide section, high vegetation is shown in dark green (72%), low vegetation in light green (22%), and the built-up area in brown (6%). Here it becomes apparent that there is a relatively large amount of high vegetation around the river, which is a sign of a relatively good and undisturbed ecological status. However, in the northeast region, urbanization has reached the river corridor and the presence of low vegetation in these areas can be explained by cutting down of trees by the neighboring population for fire and construction wood.

If all ecosystem status categories are compared with each other, it is also noticeable that the respectively better rated locations are congruent with each other. The same can be seen for both the moderately and poorly rated sites.

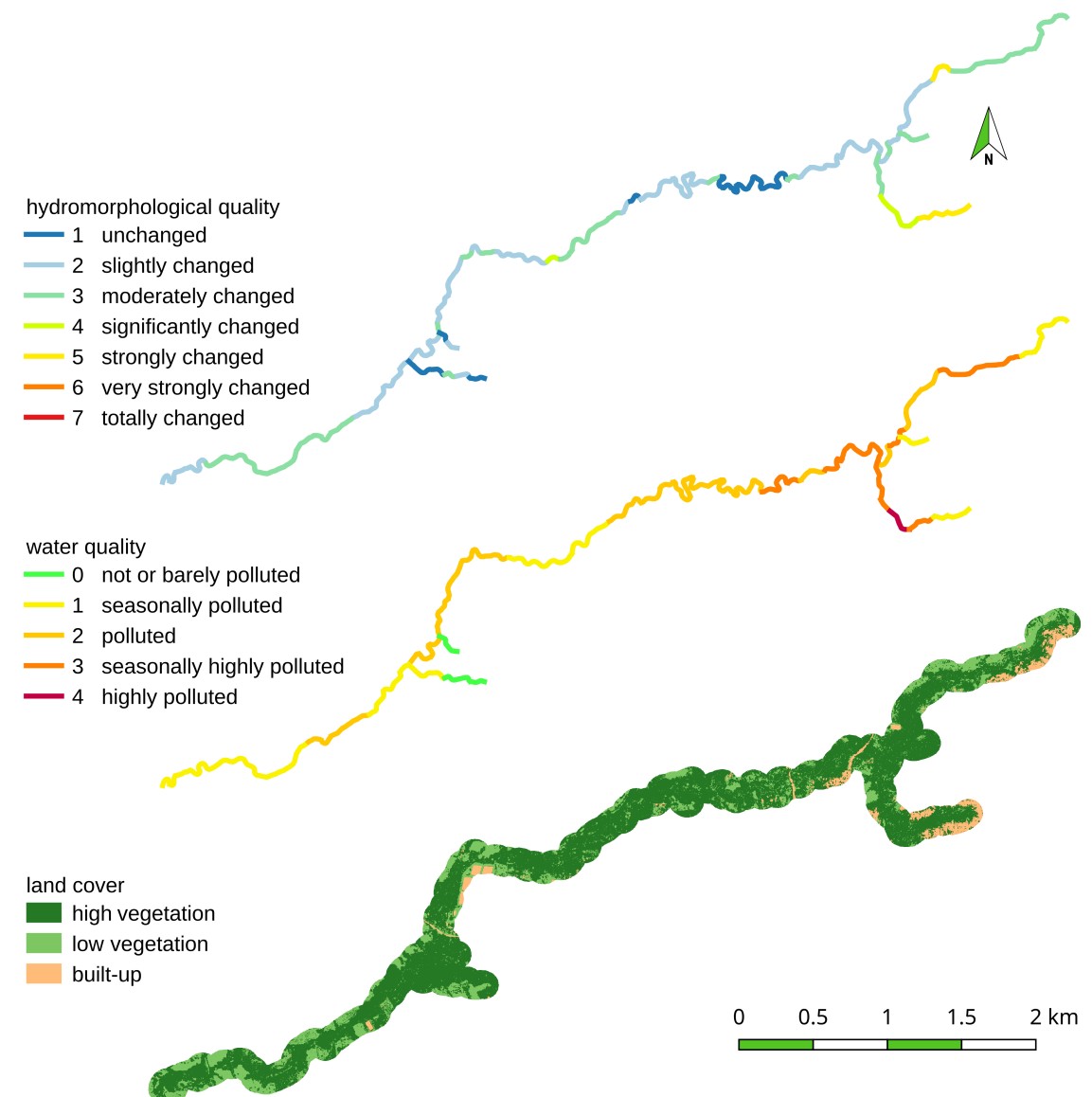

**Figure 4.** Assessment results for the three ecosystem status categories of the Pochote River.

### 3.2. Ecosystem Service Potential of the Pochote River

Based on the previous considerations and calculations, maps illustrating the potential of individual river sections to provide URES were generated (Figure 5). The potential is expressed as a percentage of a theoretically possible potential for the entire study area (131 river sections). In this specific case of the Pochote River it can be seen that for the selected ecosystem services all have a relatively good potential. However, a closer look at the detailed spatial mapping of URES per river section reveals interesting insights. River sections located very close to built-up areas or where built-up areas invade the river corridor coincide with river sections of high water pollution levels and low hydromorphological quality. This combination of bad ecosystem status in all three evaluated categories results in the lowest URES potential of all river sections. This is the case for river sections of the southern and northern river branches of the source area (URES potential of only 30–40% to 40–50% for (a) and (d), 50–60% to 60–70% for (b) and (c)).

The lowest hydromorphological quality of river sections which are surrounded by urbanized areas (river sections of the southern river branch of the source area) seems very reasonable because hydromorphological quality is mostly impaired by hydraulic stress caused by storm water drainage from built-up areas and sediment input (often construction material but also solid waste) from these

areas. Other sections with low hydromorphological quality are a result of constructive changes to the river bed (e.g., narrowing of the river bed, weirs, and bridges). These sections are located in the northern river branch of the source area and at the 90° south turn of the river when it enters the last third of the study area.

Lower quality of land cover of the river corridor and resulting lower URES potential of URES only related to this ecosystem status category, group (b) in Figure 5, is a result of built-up areas invading the river corridor or streets crossing the river (both can be observed in Figure 4). Here it needs to be stressed that the URES potential of this group is solely impaired by the presence of humans (built-up are mostly for housing) while for this group of URES (Noise attenuation & Hydrological cycle and water flow regulation & Regulation of temperature and humidity, including ventilation, and transpiration) proximity of beneficiaries to the service provision is specifically important to experience them.

Poor water quality partly originates from direct household waste water discharges where urban areas surround river sections; however, its impact on URES potential persists to river sections further downstream because water quality improves only to a low degree because of the assumed natural cleaning or when clean water of natural wells enters the river and dilutes water contamination. This can be observed in URES potential dependent on water quality at river sections of the middle course of the river before URES potential is improved by higher hydromorphological quality of a dynamically meandering part of the river (Figure 5). Additionally, poor performing waste water treatment plants as centralized facilities built downstream of settlement areas impair water quality and related URES at points further away from built-up areas. This can be observed in the deteriorating URES groups (a), (c) and (d) at the 90° south turn of the river when it enters the last third of the study area in Figure 5.

River sections with very high potential to provide URES (> 80%) are located at the confluence of the three river branches in the source area, along the strongly meandering middle course, and in the downstream area where two tributaries join the main river. The tributaries form part of the botanic garden, a protected nature reserve, where all three ecosystem categories are in a very good status. Along the middle course very good hydromorphological quality and presence of high vegetation land cover result in high URES potential. At the confluence of the three river branches in the source area, very good land cover quality is the reason for high URES potential, specifically of group (b) (noise attenuation & Hydrological cycle and water flow regulation & Regulation of temperature and humidity, including ventilation, and transpiration).

The variability in URES potential of different river sections indicates the influence of the different ecosystem status categories. Except for URES of group (b), all URES depend on a combination of ecosystem status categories. For the selected regulating URES, land cover and hydromorphology are at least equally as important as water quality. Since the considered river still preserves a good status of land cover and hydromorphology, URES potential is in most parts fairly high. If directly water quality related provisioning URES, e.g., surface water for drinking, were considered in this study, URES potential would very low for most river sections.

While Figure 5 shows a spatial distribution, Figure 6 represents distributive characteristics (maxima, minima, outliers, mean and median values, standard deviation) of all river sections as box plots for the assessed URES.

Since several URES are linked in the same way to the three ecosystem status categories, their statistical distributions are the same. This is revealed in the box plots in Figure 6.

As can be seen, there are outliers of lower URES potential for all URES. These outliers represent river sections close to built-up areas which are suffering from urban contamination and loss vegetative cover. The outliers for noise attenuation, hydrological regulation and regulation of temperature/humidity are most pronounced. Here the impact of a severe land cover loss is the reason for a low URES potential. However, for the same URES the majority of river sections have an URES potential above 90% highlighting that large parts of the river corridor still preserve a high amount of high vegetation and low or no urban impact so far. Moreover, there are individual river sections with a potential above 90% for all URES. The river has the lowest median potential in the provision of the URES 'Bio-remediation',

'Filtration' and 'Regulation of Chemical Condition'. The absolute lowest potential of a URES for all river sections is related to the provision of 'Regulation of Chemical Condition', with 40% resulting from poor water quality and altered land cover, in addition to a strongly changed hydromorphology.

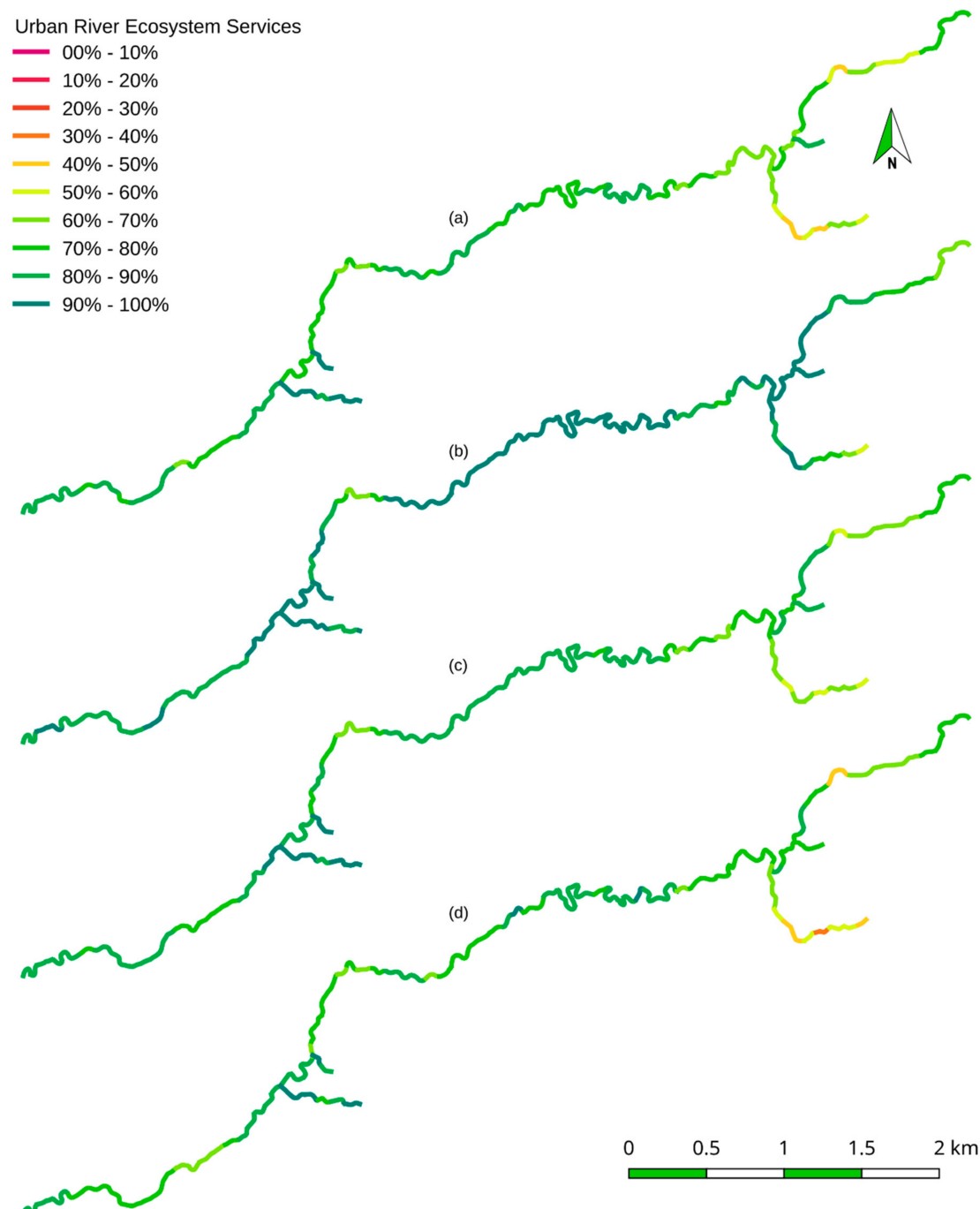

**Figure 5.** Maps of potential to provide Urban River Ecosystem Service for all sections of the Pochote River: (**a**) Bio-remediation by micro-organisms, algae, plants, and animals & Filtration/sequestration/storage/accumulation by micro-organisms, algae, plants, and animals; (**b**) Noise attenuation & Hydrological cycle and water flow regulation (Including flood control and coastal protection) & Regulation of temperature and humidity, including ventilation, and transpiration; (**c**) Maintaining nursery populations and habitats (Including gene pool protection); (**d**) Regulation of the chemical condition of freshwaters by living processes.

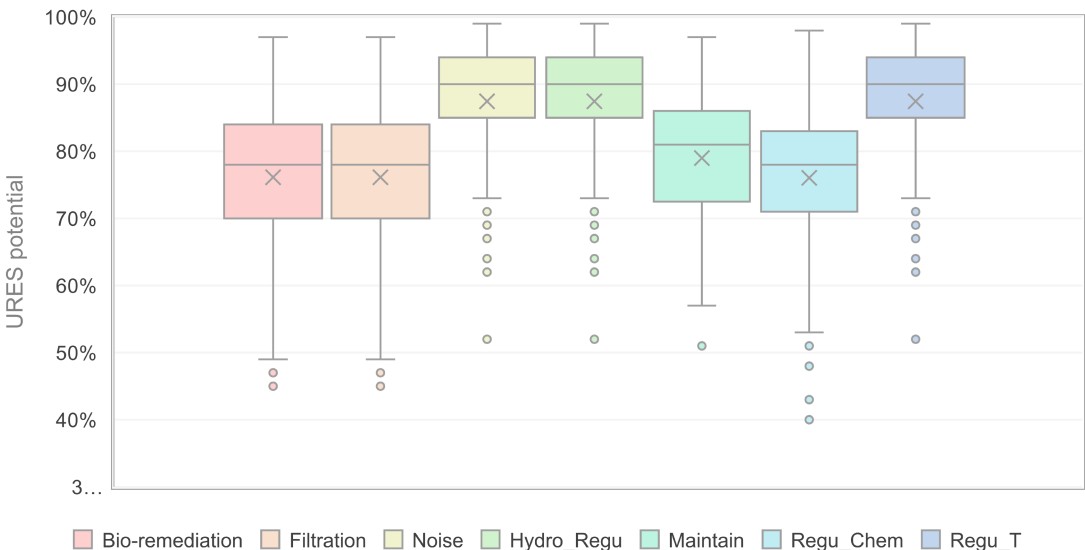

**Figure 6.** Box plots of Urban River Ecosystem Service potential of the all sections of the Pochote River: 'Bio-remediation' stands for 'Bio-remediation by micro-organisms, algae, plants, and animals'; 'Filtration' means 'Filtration/sequestration/storage/accumulation by micro-organisms, algae, plants, and animals'; 'Noise' represents 'Noise attenuation'; 'Hydro_Regu' short for 'Hydrological cycle and water flow regulation'; 'Maintain' refers to 'Maintaining nursery populations and habitats'; 'Regu_Chem' signifies 'Regulation of the chemical condition of freshwaters by living processes'; 'Regu_T' equals 'Regulation of temperature and humidity, including ventilation and transpiration'.

## 4. Discussion

To introduce a low-cost and easy-to-use methodology for assessing the potential of urban rivers and riparian areas to provide ecosystem services was the main objective of this study. The methodology is based on the use of a combination of easily obtained field information and readily accessible remote-sensing data. The following discussion will focus on the representativeness of the data used as well as the advantages and disadvantages of the developed methodology. Possibilities for improvements in data collection, use of data, and application of the methodology are considered. The established link between revealed ecosystem status data and urban ecosystem service potential will also be critically discussed. This represents the basis for the interpretation and discussion of the results. Finally, future research directions are highlighted.

*4.1. Potential of the Pochote River to Provide Ecosystem Services and Comparison with Other Assessment Methods*

Although the selected URES are in relatively good condition, there is still potential for improvement. Looking at the ES status maps (ecosystem status for the different categories), it is noticeable that the biggest negative influence on the URES, in the case of the Pochote River, is water quality. This indicates that there is still great potential for improvement of the river, as it is relatively easier and quicker to improve water quality, in comparison to restoring riparian vegetation/reverse surface sealing and the poor hydromophological quality. Even if the river is in general in a relatively good condition, it is still relatively variable over the entire route. This results in different potentials, which is clearly evident from the ecosystem services status maps. Therefore, it would be possible to use the area in the northeast preferably for local recreation. In contrast, the area in the southwest where the two tributaries are still very close to nature would be worthwhile, returning it completely to nature in order to use it for educational purposes or to promote natural cleaning. In addition to the location of the hotspots, the general picture of the quality of the URES shows that there are only a few outliers downwards. The fact that the median of each of the URES is above the mean value also confirms the

relatively good quality of the URES considered. It should also be noted that the median for all URES considered is between 78% and 90% and thus relatively high.

Lüke and Hack [39] carried out a comparative study with commonly used models to assess hydrological ecosystem services of the larger Chiquito River Basin of which the Pochote River is part of. The study compares data requirements and model performance of the Soil Water Assessment Tool (SWAT; SWAT Development Team, Temple, TX, USA), a typical hydrological model, and the RIOS model (Natural Capital Project Development Team, Stanford, CA, USA), a specialized ecosystem service model, to model provision and demand of hydrological ecosystem services within the river basin. The urban area of León is considered in both models as built-up, impacting the provision of ecosystem services. In the RIOS model, the demand for ecosystem services of the city's population is also considered. Compared to our study, the spatial scale of assessment is much larger, the river corridor is not considered as specific provisioning area of ecosystem services, and the focus is on quantification of supply and demand of services instead on the potential to provide services for urban populations. Both studies are rather complementary. Besides this study no other studies of this river basin have been published.

The use of models, traditional hydrological or specific ecosystem service models, for the assessment of freshwater ecosystem services is very common. As pointed out in the study of Lüke and Hack [39], data requirements and spatial resolution can vary a lot. Vigerstol and Aukema [54] stress that traditional hydrological tools provide more detail whereas ecosystem services tools tend to be more accessible to non-experts and can provide a good general picture of these ecosystem services. However, the compared commonly used models a limited to a spatial resolution above 30 m per pixel. Thus, these models are not developed for a detailed study of the services provided by urban river corridors as it is the case of our methodology.

Contrary to hydrological studies, ecosystem services of urban areas are often assessed at smaller spatial scales as a meta study of Schwarz et al. [4] documents. Although the very small scale of these assessments allows to identify the diversity of ecosystem service provision in an urban context, processes connected to larger spatial scales, e.g., regulation of hydrological flows or bio-remediation, are not considered.

Our study is regarding the spatial resolution and consideration of ecosystem services connecting different spatial scale in between of the study of Lüke and Hack [39] and Schwarz et al. [4]. This makes it possible to consider the "meso-scale" of an urban river corridor. We argue that this spatial scale in an urban river context is specifically relevant for local policy making as it can be easily related to urban planning, e.g., development of residential areas or street networks, and to specific development issues such as flooding, water contamination or loss of riparian habitat.

The way we translate ecosystem status to URES is through a weighted ecosystem service provision matrix. Typically, such matrices link land cover types, derived from satellite imagery or aerial photographs, with ecosystem service potential [53]. As in our ecosystem status category "land cover", lower vegetation (e.g., pasture) usually has lower potential to provide regulating services than higher vegetation (e.g., forest) [32]. The weights of the matrix are commonly provided by experts [53], just as in our case.

Shuker et al. [55] developed a set of tools to assess and communicate the biophysical condition of river and riparian habitat in urban catchments. For the assessment, information collected using the Urban River Survey, a habitat survey designed for application to 500 m stretches of urban river corridor, and comprise a series of aggregate indices, three classifications relating to the materials, habitat and vegetation characteristics of urban river stretches, which contribute to an overall score, the Stretch Habitat Quality Index [55]. The tool set was applied to tributaries of the River Thames within London. While the tool of Shuker et al. uses a variety of indicators related to hydromorphology, riparian land cover (50 m width on both sides), and water quality similar to our approach, it is based on field visits alone, and with 500 m stretches, it has a much coarser spatial resolution. The coarser spatial resolution is possibly a result of a much more detailed and expert-guided assessment of indicators.

In contrast, our approach, which is designed for a developing country context with pronounced resource constraints, is based on easily collectable field data and freely available satellite imagery. This information is combined to assess the potential of URES in order to provide relevant information for decision-making.

Other studies, such as that of Lundy and Wade [56], categorize urban water bodies with regard to their degree of naturalness (e.g., channelized water courses, natural and restored water courses), without a specific assessment of the ecosystem status, and related them to the provision of ecosystem services. This procedure seems useful to stress the benefits of urban rivers, but it is less suitable to guide decision-makers to take specific protective measure for river section where service provision is at risk.

The combination of land cover, water quality and hydromorphology in a three-dimensional service potential matrix is particularly new. With the combined use of land cover/land use-based and field-based indicators, the limitations of using only land-cover and land-use-based indicators identified by scholars of the field [7] are overcome. Our approach preserves the advantage of matrix-based assessments in illustrating complex service provision in an easily accessible manner, e.g., in policy-making context, while including additional ecosystem status categories of relevance for urban rivers.

Rivers in urban areas of developing countries are often threatened to become part of the sewer and storm water drainage system of cities. In this way, important URES are lost. In rapidly and unplanned urbanizing areas, rivers and their remaining corridors are often the only source of these ecosystem services. Ward and Winter [57] revealed, in a study in Cape Town, South Africa, that residents have a poor understanding of the linkages between what they do on the land and the impacts on urban rivers ('missing link'). A shift in focus from solely (often poorly maintained) technical solutions to avoid damage towards socio-ecological benefits that result from good ecological status could help to trigger better conservation of urban rivers. Our methodology intends to highlight the potential of urban rivers to provide ecosystem services with low data requirements and without specialized tools. The comparatively high spatial resolution with URES potential per 100 m river section allows policy makers to address specific conservation actions and to prioritize them. Since urban areas both impact and benefit from URES, the study results can facilitate the communication of trade-offs. Specifically, the study results bear important information for the planning of future urbanization. As the river corridor potentially provides essential URES it could be prioritized for conservation instead of further urbanization. Knowledge of the potential to provide URES allows the sustainable integration of rivers in urban landscapes for the benefit of both—ecosystems and city dwellers [56]. Preserving the riparian corridor can than increase well-being due to its positive influence on, e.g., residential quality and its recreational potential. Since river restoration is not yet fully on the agenda in developing countries, the explicit highlighting of URES adds valuable arguments to integrate sustainable management and conservation concepts to urban development planning.

Collier et al. [19] included in their study of the Capibaribe River in São Lourenço da Mata, Brazil, besides land cover and water quality information also local knowledge and perceptions based on semi-structured resident interviews focusing on their relationship to the urban river. Although this study is lacking a detailed assessment of the bio-physical conditions of the hydromorphology and riparian vegetation, it includes additional information on resident's perceptions of benefits derived from the river. It would be interesting to contrast our ecosystem status-based URES potential with perceptions of service provision and demand of local residents.

*4.2. Discussion and Recommendation of Improvements Regarding the Data and Methodology Used*

The principal idea of the methodology for the assessment of the potential to provide URES presented here, is that it should be based on either readily available or easily collectable data at low cost. Qualitative and also quantitative (e.g., in the case of land cover) information on the three considered ecosystem status categories can easily be redacted or collected at low cost through field visits and freely

available satellite images. Neither expert knowledge nor costly equipment is required for doing this. For this reason, we believe that it is a methodology that can be used by local government members or other public institutions in developing countries. For this reason, the data described in Section 2 was used and the methodology accordingly (data-driven) developed. However, the results are only as good as input data and suitability of methodology applied. These two aspects need to be discussed in detail.

Regarding the raw data on the ecological status of the river, not for every 100 m section data in form of photographs and the geo-localized photo documentation was available. Therefore, with some exceptions, a linear approximation between the evaluated 100 m sections was assumed. To advance the methodology, a photo should be taken from a specified angle at suitable intervals, e.g., every 50 or 100 m. This standardization could provide superior hydromorphological quality evaluation.

To determine the water quality, the number of data points was sufficient for this simple type of survey, as data were available for each discharge. However, with the applied visual classification, the use of a standardized color scale could improve the data quality and reduce potential subjectivity in visual color interpretation with little additional effort.

Even though the land cover scores provided relatively high resolution (ca. 1.5 m × 1.5 m geo-localized pixels), it was still a semi-automatic method. No general statements can be made about the accuracy of the new development of the land cover classification. This is due to the fact that this methodology is to a certain extent trial and error and that the application of the classification has been adjusted until its result has been considered satisfactory for the observation area. A random check of four different sections in the study area using the original Google map showed an accuracy of 95.6% to 99.8%. Therefore, it would be desirable for future applications to develop a guidance with precise settings to increase the degree of automatization and applicability of this method as well as to quantify the accuracy of this method in general.

Using Google Earth images for land cover analysis in the river corridor allows a very high-resolution classification which is of special importance for identifying very small-scale structures within a relatively narrow area. This high resolution is actually needed to recognize even smaller impacts commonly observed when urbanization occurs. The land cover classification used for this study efficiently revealed and quantified small-scale structures such as paths crossing the river corridor, individual buildings and riparian vegetation clearings. This represents a valuable contribution of remote sensing to facilitate assessments of urban river ecosystem services. Additionally, the use of Google Earth images, which are freely available for non-commercial use, as proposed in our study is an important source of information for many regions in the world where official land cover data of high resolution is often missing.

For future applications without the resource and information constraints of the study presented here, other remote sensing data, e.g., provided by other satellite missions or unmanned aerial vehicles (UAV), can be incorporated in the presented method in the same way as the Google Earth images to assess the ecosystem status concerning land cover. Other data can potentially provide more detail on land cover structures, e.g., Leaf Area Index (LAI) or vegetation stress status. UAV equipped with Light Detection And Ranging (LIDAR) or multispectral cameras could possibly provide information also to assess the other two ecosystem status categories considered in our methodology, hydromorphological quality and water quality respectively.

Thus, the data described here is of relatively good quality and/or the methods of collecting them can be easily improved. However, as soon as the transformation into the URES takes place, several assumptions are made. Since all these data can be collected quickly and inexpensively, the URES survey method is well suited for a rough analysis of the current state of the study area and should also be regarded as such. Thus, this method can be used for the purpose of searching for a suitable field of research or action. Furthermore, the ES status maps should not be regarded as a general state of ecosystem services in the study area, as only a small selection of the URES was covered. It should also be mentioned that these are only 'Regulation & Maintenance' URES. However, the underlying ecosystem status data can be extended with little effort so that statements can be made about cultural

URES. For example, the land cover map can be used to derive a value for the population density per area and its distance to the river and thus a potential for the URES 'Physical and experiential interactions with natural environment'. There may also be census data available that can replace the population density value derived from the map and further enhance the quality of the data. Data for the 'Intellectual and representative interactions with natural environment' URES can also be collected in the same way, although adjustments must be made depending on the URES. For some, the distance between schools or universities to the river leads to the desired result, while for others the survey can remain unchanged. For the schools as well as for the universities it would be even more helpful to know how many students or scientific staff they have, as this can provided quantitative information on the number of potential beneficiaries.

The degree of the relationship between ecosystem status and URES is debatable, but there definitely is a relationship. The relationships and weighting established in this publication can be interpreted as a rough estimation of the potential to provide ecosystem services. The relationship was established on a relatively detailed ES "class" level of CICES. A more detailed consideration of specific "class types", as established with CICES version 5.1. [27], within these CICES classes would imply a quantitative assessment of class types (e.g., flood volume attenuated) and would rather lead into a detailed assessment and not an analysis of potential as it was done for this study. However, a further elaboration with more detailed consideration of the ecosystem status categories could provide information to assess class types.

There is a general problem of a lack of data and indicators to assess the large variety of URES, especially in developing countries. The methodology proposed here can help to structure and formalize the assessment of the ecological status of urban rivers and their riparian corridors in a simple but still sufficiently sound manner. Already the information on the ecological status, reflected in three categories that represent different biotic and abiotic realms and characteristics of a river corridor, facilitates the identification of deterioration of ecosystems due to urbanization. This alone can guide decision-making with regard to establishing protection zones and/or regulating urban development. Linking the ecosystem status categories to URES to evaluate the potential of river sections to provide URES can then assist to stress the benefits from urban rivers in good ecological status establishing the basis to argue for the protection and sustainable uses of urban rivers. In a next step, this service provision potential could be contrasted with potential beneficiaries to identify specific supply and demand relationships [39]. At this point, other URES, including provisioning and cultural ones, may be included to identify additional benefits beyond those assed in this study and other beneficiaries. An indicator set for the assessment of cultural ecosystem service of river landscapes has recently been developed for Germany [58], which may contribute additional ecosystem status categories or links to URES, although it has not been developed specifically for urban river landscapes.

## 5. Conclusions

Rivers and their riparian zones represent often the only remaining natural-like spaces in urban areas of developing countries. In a landscape transition from rural to urban land uses, the original importance of rivers (to provide water for households, irrigation and animals) vanishes and a transformation of them into wastewater channels typically occurs. While intact semi-urban or urban rivers can have a significant regulative and recreational function, the frequently rapid and uncoordinated urbanization process poses a significant threat of losing these areas and the benefits to the population they may provide. Hence, establishing the specific category of Urban River Ecosystem Services, as done in this contribution, to stress this important ecosystem-society link seems useful. The methodology presented here allows local decision makers, e.g., city administrations or water companies, to translate data and information from field surveys combined with easily accessible remote-sensing data into three ecosystem status information. The resulting maps represent a baseline of the ecosystem status of an urban river. The subsequent weighted combination of them facilitates then the assessment of the potential of different river sections to provide specific ecosystem services to the population. This may

serve as a guidance to identify hotspots of loss and high provisioning areas of ecosystem services. The ecosystem service concept has been developed as a strong communication tool for the benefits that ecosystems provide. Especially in urban contexts, where dynamics and pressure on ecosystems are extraordinary, the highlighting of ecosystem service potential may contribute to a sustainable use and protection of vulnerable ecosystems.

**Supplementary Materials:** The following are available online at http://www.mdpi.com/2072-4292/11/14/1697/s1, Table S1: Urban river Ecosystem Services Classification according to Common International Classification of Ecosystem Services (CICES) 5.1. Author's work adapted from Haines-Young and Potschin [21] and Maes et al. [34,35].

**Author Contributions:** Conceptualization, M.R.B. and J.H.; Investigation, M.R.B. and J.H.; Methodology, M.R.B. and J.H.; Software, M.R.B.; Supervision, J.H.; Visualization, M.R.B.; Writing—original draft, M.R.B. and J.H.

**Funding:** This research was funded by the German Federal Ministry of Education and research (BMBF), grant number 01UU1704.

**Acknowledgments:** We acknowledge support from the German Research Foundation and the Open Access Publishing Fund of Technische Universität Darmstadt. Further we want to express our gratitude to the anonymous reviewers for their valuable comments on the manuscript.

**Conflicts of Interest:** The authors declare no conflict of interest.

## Appendix A

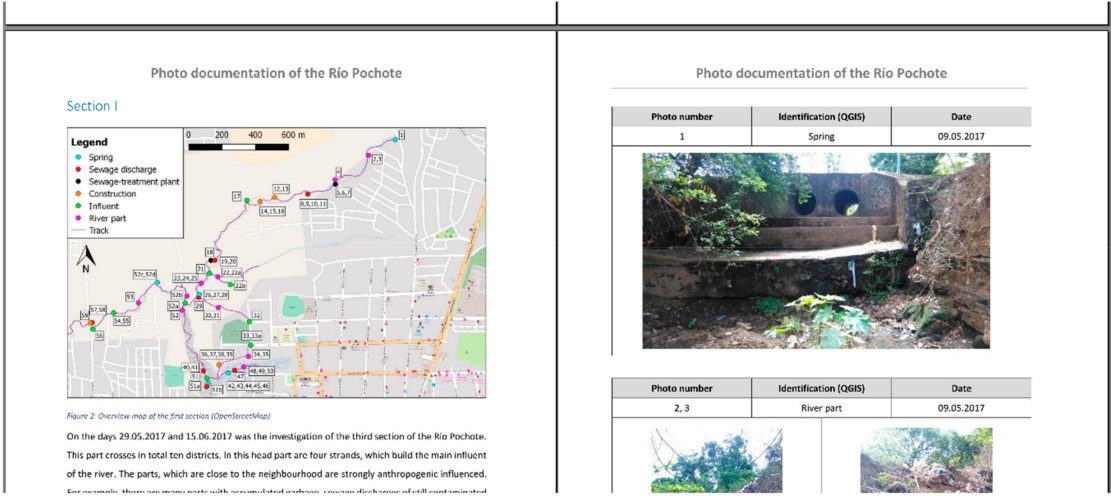

**Figure A1.** Cutout of a photo documentation of the Pochote River [40].

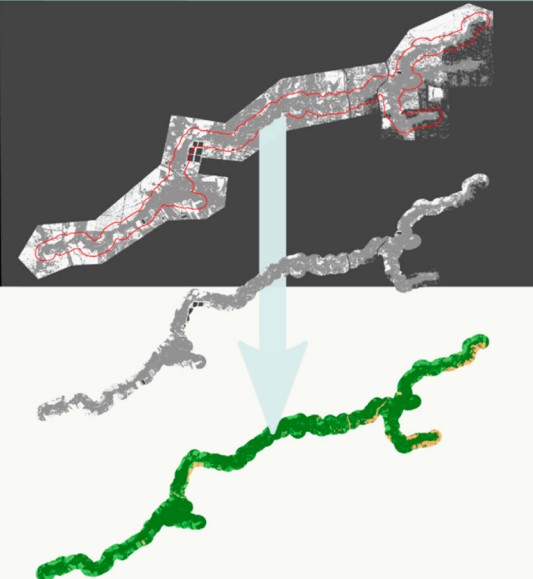

## Map Preparation Process with GIMP

**trimming**

- ➢ Open google satellite image with GIMP
1. Layer \ Trancparency \ Add Alpha Channel
2. Free Select Tool -> roughly select the study area
3. Select \ Invert
4. Layer \ Trancparency \ Treshhold Alpha...; Value: 0.0000 *

**main coloring**

5. Select by Color Tool; Mode: Add to the current selection, Treshhold: 10.0, Select by: Composite ➢ Select low vegetation
6. Free Select Tool; Mode: Subtract from the current section ➢ roughly select the city area ➢ Color the selected area in yellow ➢ deselect all **
7. Select by Color Tool ➢ Select low vegetation ➢ color the selected area pink ➢ deselect all
8. Select by Color Tool ➢ Select low build-up area ➢ color the selected area violet ➢ deselect all
9. Select by Color Tool ➢ Select the shadows
10. Free Select Tool; Mode: Intersect with the current section ➢ roughly select the city area ➢ Color it violet
11. Select by Color Tool ➢ Select the remaining shadows ➢ Color it pink ➢ deselect all ***
12. Free Select Tool; Mode: Add to the current selection ➢ Select all water treatment plants ➢ Color it pink °

**small fixes**

13. Colors \ Brightnes-Contrast...; Contrast: 127 °°
14. Select by Color Tool ➢ Select the violet, pink and yellow area
15. Select \ Invert ➢ look in comparison to the original image to which area belonging the most of the selected pixels ➢ colorize to the belonging area (here: Selected pixels inside the city = build-up area & selcted pixels outside the city = high area)

**export**

16. Image \ Mode \ Indexed; Generate optimum palette, Maximum number of colors: 3 °°°
17. File \ Export as... ➢ save as .png file ➢ click on export ➢ change pixelformat: 8 bpc GRAY *°

\* The rough study area was croped this way to be able to use the GCP-Points of the original satellite image for the geo-referencing process of the in GIMP produced .png file

\*\* This process is necessary because there are many brown roofs and streets (covered by dust) in the city.

\*\*\* Here the shadows of the build-up area and the high vegetation were added to get closer to the goal of only 3 colors.

° Due to the green light reflection of the water of the water treatment basins, the automatic part of the method incorrectly assigned

°° Set the contrast to the maximum is necessary for the following step, since slightly different gradations of each color tone were created during the coloring process (especially in transition areas).

°°° This step is necessary to eliminate the very last slight colour variations of the three different colours.

*° Format that performs excellently for georeferenced rasters in QGIS

## Georeferenced Raster with QGIS

18. Install the 'Georeferencer GDAL' plugin and the 'OpenLayers Plugin'
19. Activate with the OpenLayers Plugin the Google Satellite
20. Open the original satelite image in the georeferencer plugin
21. Use the plugin to select the same 5 points on each map (scattered over the study area)
22. Go to settings and adjust: Transformation type: Polynomial 1, Resampling method: Nearest Neighbor,
23. Save the GCP-Points
24. Open the in GIMP produced .png file and the even produced GCP-Points
25. Start the georeferencering process to produce a georeferenced QGIS raster
26. Save the raster in the needed Coordinate Reference System
27. Clip the raster file with the layer of the study area and color it as needed

**Figure A2.** Manual of producing a land cover map with the software GNU Image Manipulation Program (GIMP) and the geoinformation software QGIS.

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
