# Peer review of "A Combined Field and Remote-Sensing Based Methodology to Assess the Ecosystem Service Potential of Urban Rivers in Developing Countries"

_remotesensing, doi:10.3390/rs11141697_

Round 1
Reviewer 1 Report
I think it should be deleted in the title and the text devoloping countries,because the methodology must be valid for all similar cases to the study
Reviewer 2 Report
I have no more comments. Good work!
Reviewer 3 Report
The manuscript was significantly improved. The only small correction in English grammar is needed, e.g. in the sentence in lines: 455-458.
This manuscript is a resubmission of an earlier submission. The following is a list of the peer review reports and author responses from that submission.
Round 1
Reviewer 1 Report
The manuscript „ A combined field and remote‐sensing based methodology to assess the ecosystem service potential of urban rivers in developing countries” concerns very important aspect on an innovative methodology for a rapid and low cost assessment of the ecological status of urban rivers and riparian areas. Such results are worth to publish in Remote sensing. However, some revision is needed, primarily of the statement a clear aim of this study in Introduction, correction of the methods and the other small corrections
Detailed comments:
Introduction
Please clarify the aim of the study in the Introduction in lines: 87-97. Why did you indicate the clear aim only in Discussion – lines 517-518: “The objective of this article is to introduce a low‐cost and easy to use methodology to assess the potential of urban rivers and riparian areas to provide ecosystem services.” I think the last paragraph of Introduction and the first paragraph of Discussion should be corrected.
Material and Methods
This part of manuscript is described in a very detailed way. However, Table 2 requires corrections. Why did you classify as follows:
unchanged
slightly unchanged
moderately unchanged
significantly unchanged
strongly unchanged
very strongly unchanged
totally changed
I think in classes from 2 to 6 should be “…changed” instead of “…unchanged”. Please compare the classification with Results – Figure 4.
In lines 355 and 362 please indicate the names of [22]. …. according to Haines‐Young and Potschin [22]….
Results
Lines 491-493: “While Figure 7 shows a spatial distribution, Figure 8 represents distributive characteristics (maxima, minima, mean and median values, standard deviation) of all river sections as box plots for the assessed URES.” outliers were missed in the brackets.
Discussion
Lines 164, 359, 600: Why did you used “…an (urban) river corridor…”?
Please avoid the citing Figures in Discussion.
Small corrections of English language in the whole text is needed. The explanation of the codes: URES and CICES should only take place for the first time.
Author Response
Dear reviewer 1,
thank you very much for reviewing our manuscript. Your valuable comments and suggestions have helped us to improve our work further. We addressed all of them in our revision. Please find our responses to your comments in the file in the download section.
Best regards on behalf of the authors

Reviewer 2 Report
Deterioration of urban rivers and their riparian areas is an important issue in urban environment management. This paper presents an innovative methodology for a rapid and low cost assessment of the ecological status of urban rivers and riparian areas in developing countries under data scarce conditions. This paper is very well-written. The structure is very clear and the contributions are well highlighted. I enjoy reading the paper very much.
Some minor suggestions:
L153, Section 2.2 may be too long since all the methods are introduced in the following sub sections.
L227, a short sentence can be added to explain why 100 m is chosen to divide river sections?
L228-233, how to determine different hydromorphological quality classes from satellite imagery?
Figure 7, I cannot see 0-10% and 10-20% colors in the figure.
Figure A1, the quality of this figure is not good compared to other well-illustrated figures of the paper.
Author Response
Dear reviewer 2,
thank you very much for reviewing our manuscript. Your valuable comments and suggestions have helped us to improve our work further. We addressed all of them in our revision. Please find our responses to your comments in the file in the download section.
Best regards on behalf of the authors

Reviewer 3 Report
This manuscript firstly defined several categories for Urban River Ecosystem Services (URES), then chose three ecosystem status (i.e. hydromorphology, water quality, and land cover) that can be easily obtained from field survey and remote sensing data, and finally linked them to generate several ecosystem services status maps for the Pochote River, in the City of Leon, Nicaragua. I have several concerns on the manuscript which don’t allow me to recommend it for publication in the journal of Remote Sensing.
General comments:
1. Lack of contributions to remote sensing community. Although a land cover map used in this study is obtained from a satellite image, it is just a simple application of the existing remote sensing techniques. Since there is no any comparison between the results with and without remote sensing data, it is not clear what are the advantages of the remote sensing. Therefore, it can be considered that this manuscript is not suitable to be published in a remote sensing journal and it is better to choose other journals relating ecosystem services.
2. The authors declared that they present “an innovative methodology for a rapid and low-cost assessment of the ecological status of urban rivers and riparian areas”. However, it is not clear what are the innovations in the developed method due to lack of comparisons with other existing methods.
3. Although there is no problem to follow the manuscript, the clarity of presentation and English grammar (e.g., "see Figure 5-7" in lines 528-529 should be “Figures 5-7”, “This results” in line 533 should be “These results”, and so on) of the manuscript still need additional work.
4. The quality of almost figures and tables should be improved.
Specific comments:
Introduction: please make clear what are the research objectives of this manuscript in the last paragraph.
Lines 119-121, what’s meaning of the numbers (i.e. (1), (2), and (3)) in this sentence? It is confused.
Figure 1, this figure was not mentioned in Section 2.1. In addition, the colors in background of the figure represent what?
Figure 2, this figure is difficult to see and understand. A general flowchart for the methodology maybe better. What’s meaning of the colors in this figure?
Line 226, “(Figure 4)” should be removed because it is a result.
Table 1, this table should be improved according to Author Instruction. Generally, only three horizontal lines are needed.
Lines 271-282. Explanations for variables in Eq. (1) and Eq. (2) are confused. For example, “WQj” should be “water quality value of section j”, “WQ” is a variable in Eq. (2), not in Eq. (1), “WQdi” should be “WQdi, i, j”, …
Table 2, this table should also be improved according to Author Instruction.
Line 377, “and land cover an indicator for…” should be “and land cover is an indicator for…”, right?
Lines 377-379. This is a confused sentence. Do you mean “hqk:wqk:lck=1:1:2”?
Line 392, this is a confused sentence. Please rewrite.
Table 3, this table should also be improved according to Author Instruction. Colors are no need.
Line 413, “combination” should be “contribution”, right?
Lines 414-424, Explanations for variables in Eq. (3) are confused. Please explain each variable with its footnote.
Line 450, “northwest” should be “northeast”, right?
Figure 5, it is difficult to see yellow color for “seasonally polluted” category.
Figure 7, (a), (b), (c) and (d) should have the same size.
Figure 8, please provide the name for Y axis. Colors are no need.
Lines 528-529, “see Figure 5-7” should be “see Figures 4-7” or “see Figures 4-6”, right?
Conclusions, this section should be revised as it is not truly to conclude something obtained from the results in the manuscript.
Author Response
Dear reviewer,
we thank you very much for reviewing our manuscript. Your valuable comments and suggestion have helped us to further improve it. We addressed all of your comments in our revised manuscript to the extend possible in order to improve the quality of the content and presentation of our work. Please find our point-by-point response and particular information on how we addressed each comment in the document attached.
Thank you again for your support.
With best regards on behalf of the authors,
Jochen Hack

Reviewer 4 Report
There are indicators as LAI that are dynamics, changes in the time or season and growth of each year. In addition, the values of LAI depends of type of vegetation and the water quality depends on runoff regime that suggest the self-depuration capacity of river. As well could be take account of what is the influence of channel size, slope, and size distribution of particles in the potential ecosystems services. May be change the title thinking about the potential current
Author Response
Dear reviewer 4,
thank you very much for reviewing our manuscript. Your valuable comments and suggestions have helped us to improve our work further. We addressed all of them in our revision. An extensive editing of language and style was carried out. Please find our responses to your comments in the file in the download section.
Best regards on behalf of the authors

Round 2
Reviewer 3 Report
The authors have addressed all my comments.